# The Role of Pain Inflexibility and Acceptance among Headache and Temporomandibular Disorders Patients

**DOI:** 10.3390/ijerph19137974

**Published:** 2022-06-29

**Authors:** Vanessa Marcelino, Maria Paço, Andreia Dias, Vera Almeida, José Carlos Rocha, Rui Azevedo, Miguel Alves-Ferreira, Carolina Lemos, Teresa Pinho

**Affiliations:** 1UNIPRO—Oral Pathology and Rehabilitation Research Unit, University Institute of Health Sciences (IUCS), CESPU, 4585-116 Gandra, Portugal; vanessa.k.marcelino@gmail.com (V.M.); maria.paco@ipsn.cespu.pt (M.P.); vera.marg@gmail.com (V.A.); jose.ferreirinha.rocha@gmail.com (J.C.R.); rui.azevedo@iucs.cespu.pt (R.A.); 2UnIGENe, IBMC—Institute for Molecular and Cell Biology, i3S—Instituto de Investigação e Inovação em Saúde, Universidade do Porto, 4200-135 Porto, Portugal; andreia.dias@ibmc.up.pt (A.D.); miguel.ferreira@ibmc.up.pt (M.A.-F.); clclemos@ibmc.up.pt (C.L.)

**Keywords:** chronic pain, headache, pain acceptance pain inflexibility, psychosocial variables, Temporomandibular joint syndrome

## Abstract

Temporomandibular disorders (TMD) and headache are complex. This study aims to assess the association between TMD, headache, and psychological dimensions such as psychological inflexibility and pain acceptance. The sample consisted of 120 participants following a non-probabilistic convenience sampling strategy through a direct invitation to the patients attending our facilities and their relatives (*n* = 61 diagnosed with headache, *n* = 34 diagnosed with TMD-headache, *n* = 25 control group). Diagnostic Criteria for Temporomandibular Disorders (DC-TMD), International Classification of Headache Disorders (ICHD-3 beta version), Chronic Pain Acceptance Questionnaire (CPAQ-8), and Psychological Inflexibility in Pain Scale (PIPS) were used as assessment tools. One-way ANOVA, multiple regression analysis (MRA), and the Johnson-Neyman approach were run by IBM SPSS, version 27 (IBM^®^ Company, Chicago, IL, USA). The significance level was 0.05. One third of our sample presented with headache with TMD. Females were predominant. Males with headache, no systemic disease, less pain severity but higher frequency, living longer with the disease and having sensitive changes, showed higher pain acceptance. When headache occurs with TMD, women with higher education, no headache family history, less pain, and no motor changes showed higher pain acceptance. Patients with both conditions are more liable to have chronic pain and pain inflexibility. Pain intensity and willingness explain 50% of the psychological inflexibility in the headache group. In our sample, individuals suffering from both conditions show greater pain inflexibility, implicating more vivid suffering experiences, leading to altered daily decisions and actions. However, further studies are needed to highlight this possible association.

## 1. Introduction

Temporomandandibular disorders (TMDs) and headache are complex phenomena; they represent an important public health problem [1,2,3,4] and are influenced by multiple factors, including biological, psychological, social, and behavioral factors [5]. TMDs are the second most common musculoskeletal condition [6] and headaches are a frequent medical complaint [7] that rank third in terms of cost among neurological disorders [8]. These clinical conditions are closely related, with the prevalence of headache in the population with TMD varying between 48% and 85%, while in the general population the prevalence of headache is around 45% [9,10]. They are very often found together, with patients diagnosed with both conditions reporting significantly higher levels of pain and disability compared to patients with only TMD [1,11]. The comorbidity between TMD and headache disorders has been explained by the clinical and pathophysiological features that seem to overlap, indicating the existence of common mechanisms [12]. The shared neural pathways for pain transmission, modulation, and perception have been described, as well as the central sensitization mechanisms [13,14,15]. These comorbidities are well recognized and can become chronic and disabling musculoskeletal and neurological conditions [16]. There are several effects of chronic pain on physical, psychological, and social functioning. Highlighting the impact of patients’ cognitive content on exacerbation or pain maintenance is of extreme importance, as well as pain-related health care demand and response treatments [17]. Within the psychological domain, evidence has shown that patients with musculoskeletal pain present an altered psychological function, including increased levels of emotional distress, somatic awareness, psychosocial stress, and maladaptive coping [17,18,19,20]. Although psychological symptomatology is often interpreted as a consequence of chronic pain, prospective studies suggest that pre-morbid psychological dysfunction represents a risk factor for the future development of chronic pain [21,22,23]. Two main dimensions of pain that have been described in the literature are Pain Acceptance (PA) and Psychological Inflexibility. The first is defined as the tendency to experience chronic pain without the need to reduce, avoid, or make any attempt to change it [24]; it thus entails a readiness to experience or suffer pain sensations (pain willingness) as well as a readiness to engage in activities despite discomfort (activity engagement), especially activities that are in line with one’s overall life objectives and values [25,26]. The second is defined as the inability to take value-based actions—experiential avoidance—in the presence of unwanted thoughts, feelings, or bodily symptoms, and is associated with negative health outcomes, including depression and anxiety [27,28,29,30]. For the above reasons, the aim of this study is: (a) to verify the psychological inflexibility in pain and pain acceptance between studied groups; (b) to know the variables that contribute to explaining psychological inflexibility in pain and pain acceptance, in headache sufferers and both headache and TMD sufferers.

## 2. Methods

### 2.1. Study Design and Participants

An observational, cross-sectional, and analytical study was carried out at two major health teaching and research facilities in Northern Portugal: CESPU (Cooperativa de Ensino Superior Politécnico e Universitário) and i3S (Institute of Health Research and Innovation of the University of Porto), between November 2019 and March 2020.

A non-probabilistic convenience sampling method was applied, through a direct invitation to the patients attending these facilities and their relatives, with 120 participants (*n* = 61 diagnosed with headache, *n* = 34 diagnosed with TMD-headache, *n* = 25 control group) being enrolled in the study. To be included in the study the volunteers had to have a diagnosis of TMD and/or headache or should be a direct family member of these individuals. Participants were excluded if they presented rheumatic or metabolic syndromes.

Ethical approval was obtained from the ethical committee of the University Institute of Health Sciences (register number 26/CE-IUCS/2020). All participants provided informed consent, and all procedures were conducted according to the Declaration of Helsinki.

### 2.2. Procedures

Before initiating the main study, a pilot study was performed to evaluate intra and inter-observer reliability, with 30 participants that had the same characteristics as the main study sample and that were not included in the final sample analysis. For this, 3 examiners performed the evaluation procedure independently in 2 moments (separated by a wash-out period of one week). Most of the Kappa values are between 0.65 and 1.0, which show a substantial level of agreement between intra and inter-examiner agreement, providing consistency in the results of the data survey carried out.

After assessing eligibility criteria, participants were first asked about demographic characteristics. They then were evaluated considering temporomandibular disorder diagnosis (present or absent through Diagnostic Criteria for Temporomandibular Disorders—DC/TMD) and headache diagnosis (International Classification of Headache 3—ICDH 3), psychological pain inflexibility (Psychological Pain Inflexibility Scale—PIPS), and acceptance of chronic pain (Chronic Pain Acceptance Questionnaire—CPAQ-8).

The examination took place at CESPU and i3S, and was performed by trained health professionals following the well-established and described procedures for the diagnosis of TMD—DC/TMD (original version from Schiffman et al., 2014 [31] and Portuguese version from Orbach et al. [32]) and headache—ICDH-3 [33]. The psychosocial variables were evaluated by the previously mentioned questionnaires and scales, chosen for their psychometric characteristics and availability in the Portuguese language and were taken during an interview with the patients [33]. 

Chronic Pain Acceptance Questionnaire (CPAQ-8)—original version from Rovner et al., 2014 [34], and Portuguese version from Costa et al., 2014 [35].

CPAQ-8 is a reduced version of the CPAQ-20, consisting of a scale of eight items with two subscales: “Activities engagement” (the degree to which the person engages in activities with present pain) (items 1, 2, 3, and 6) and “Pain willingness” (the degree to which the person abstains from trying to avoid or control experiences) (items 4, 5, 7, and 8). It is a Likert-type scale that ranges from 0 (never true) to 6 (always true), in which the higher scores indicate greater involvement in activities and more disposition for pain. In the study by Rovner et al., 2014 [34] the internal consistency of the scale was 0.80, whereas in the Portuguese version it was 0.95 [35].

The Psychological Inflexibility in Pain Scale (PIPS)—original version is from Wicksell et al. 2008 [36], and the Portuguese version is from Galhardo, A. et al., 2018 [37].

The Psychological Inflexibility in Pain Scale is a reduced version with 12 items, with two subscales: “Avoidance” (measures the behavioral tendency to move away from planned and valued activities and social participation in response to pain) (items 1, 2, 4, 5, 7, 8, 10, and 11) and “Cognitive Fusion” (measures the crossing of thoughts related to pain and real experiences) (items 3, 6, 9, and 12), with a Likert scale ranging from 1 (never true) to 7 (always true), in which higher scores indicate greater psychological inflexibility. In the original study by Wicksell et al. (2010), the internal consistency of this scale was 0.87 [36], while in the Portuguese version it was 0.94 [37].

### 2.3. Data Analysis

Descriptive statistics (mean, standard deviation, minimum, maximum, frequencies, and percentage) were used to understand participants’ sociodemographic and clinical characteristics and to calculate dependent variable values according to group variables. The chi-squared test was used to assess the significance of the different distributions. Also, a one-way ANOVA (analysis of variance) was used to compare the means of three independent groups (control, headache, and TMD-headache) to determine if there is a statistically significant difference between the corresponding population means. Several hierarchical multiple regression analyses (MRAs) were carried out to assess the relationship between a dependent variable (psychological inflexibility in pain or pain acceptance) and various independent variables (sociodemographic and/or clinical variables). Finally, to analyze the moderator role of illness years in the relationship between pain willingness and psychological inflexibility in pain in the headache group, the moderation assumptions were tested and fulfilled using Macro Process for SPSS, version 3.5, and the Johnson-Neyman technique; this technique determines the transition point in which the pain duration variable was significant enough to notice a difference in the relationship between pain willingness and psychological inflexibility in pain. All analyses were performed using the statistical analysis program IBM SPSS, version 27 (IBM^®^ Company, Chicago, IL, USA). The significance level was set at 0.05.

## 3. Results

### 3.1. Sample’s Sociodemographic and Clinical Characterization

Our sample was mainly characterized by females (66.7%) in a relationship (59.2%) with no higher education (65%). The mean age was 46 years old (*SD* ± 16.64) (Table 1). 

When it comes to the clinical variables, half of our sample presented a systemic disease (52.5%). From the systemic diseases reported, diabetes (64.52%) was the most prevalent among the participants, followed by depression (4.84%) and others (that included epilepsy, cardiovascular diseases, cancer, dyslipidemia, hypertension, asthma, and Crohn’s Disease). Concerning medication use, 66.7% of the participants mentioned taking medication, where 3 participants (3.75%) referred taking NSAIDs, 2 participants (1.25%) myorelaxants, 14 participants (16.25%) benzodiazepines, and 63 participants (78.75%) were taking other types of medication.

It is important to notice that less than half of the sample went to the doctor because of headache (35.8%). Half of the sample has presented this disease for less than 20 years (51.7%); almost a third of the sample indicates suffering from headache associated with anxiety, stress, and nervousness (29.2%) (Table 1). Concerning pain severity, its modalities are evenly distributed (light = 30.8%, moderate = 23.3%, and severe = 26.7%), presenting a mean intensity of 5.82 (*SD* ± 3.45) values in 10. Most of the sample has presented permanent pain for 21 years (*SD* ± 16.61), on average. More than half of the sample (56.7%) did not have any medical exams to present (laboratory analyses and/or radiological results) nor have done had any preventive treatment (beta-blockers, anti-epileptics, antidepressants). However, most of them referred to having previous treatment (64.2%), using NSAIDs, anxiolytics/miorelaxants, and analgesics, among other medicines. Few participants reported sensory, motor, or visual changes, the latter being far more reported than the previous. Almost half of the sample (44.2%) consider that anxiety/nervousness causes headache; on the other hand, a fourth of the sample (25.8%) believe those factors to aggravate headache. Most of the sample had prior headache diagnosis (79.2%).

Regarding two of our study groups (headache and TMD-headache), we verified a predominance of symptomatic females (65.6% and 79.4%, respectively).

Sociodemographic variables did not show statistically significant differences between groups. However, differences between groups were statistically significant in almost all clinical variables, except for systemic diseases, medication, and years of pain (Table 1).

### 3.2. Psychological Differences between Groups

The means of psychological variables were also compared (Table 2). Statistically significant differences were found in psychological inflexibility in pain (total) (*p* = 0.002), avoidance of pain (*p* = 0.016), and pain cognitive fusion (*p* = 0.001). The differences are significant between the control and the TMD-headache groups, and between the headache and the TMD-headache groups. The differences were not statistically significant between the control and headache groups (Table 2). The TMD-headache group is the one with the highest values in the three studied pain dimensions. No differences were found between groups in chronic pain acceptance (total), activity engagement, and pain willingness (Table 2). 

### 3.3. Variables That Contribute to Explaining Psychological Inflexibility in Pain-Headache and TMD-Headache Groups

Pain intensity and pain willingness explain 50% of the psychological inflexibility in pain variance in the headache group (Table 3). We can see that higher pain intensity and lower availability to pain contribute to increasing psychological inflexibility in pain. Pain intensity and anxiety/nervousness that cause headache explain 44% of the psychological inflexibility in pain variance in the TMD-headache group (Table 4).

### 3.4. Variables That Contribute to Explaining Pain Acceptance in Headache and TMD-Headache Groups

Gender, systemic disease, pain severity, pain frequency, illness years, and sensitive changes explain 28% of pain acceptance variance in the headache group (Table 5). Meaning that being a male, not having a systemic disease, feeling less pain severity, having a higher frequency of pain, living more years with the disease, and having sensitive changes, contributes to higher acceptance of pain. 

Regarding the TMD-headache group, 27% of the pain acceptance variance is explained by gender, education, family with history of headache, pain frequency, and motor changes (Table 6). Indicating that in this group, being a female, having more education, having no family history of headache, having less frequency of pain, and having no motor changes contribute to a higher acceptance of pain.

### 3.5. Illness Years as a Moderator between Pain Willingness and Psychological Inflexibility in Pain in Headache Group

The model that tested the moderator role of illness years in the relationship between pain willingness and psychological inflexibility in pain in headache group was significant [*F*(3, 57) = 16.14, *p* < 0.001; β = −0.42, 95% CI (−0.079, −0.058), *t* = −2.32, *p* = 0.024], explaining 46% of variance. Hence, there was a negative relationship between pain willingness and psychological inflexibility in pain in the headache group when the illness years variable was high (β = −1.95, 95% CI [−2.58, −1.32], *t* = −6.23, *p* < 0.001). The Johnson-Neyman technique showed that pain willingness was significantly correlated with psychological inflexibility in pain in the headache group when the standardized value of illness years was 0.63 above the mean (β = −0.75, *p* < 0.050), and this was true for 93.44% of the sample (Figure 1).

## 4. Discussion

For decades, researchers have studied the connection between headache and TMD [10,38,39,40,41,42]. These diseases are related to central nervous system neurobiological, physiological, and morphological alterations [43,44]. TMD and headache comorbidity enhances the difficulty of diagnosing both TMD and headache (possibly leading to misdiagnosis) [45,46] and increases the difficulty of managing both TMD [14] and headache [41]. 

Several epidemiological and clinical studies, conducted in different countries, showed that headache and TMD are highly prevalent conditions in the population [6,47]. One-third of our sample presented both conditions simultaneously and half presented only headache complaints.

### 4.1. Psychological Inflexibility in Chronic Pain 

TMD-headache comorbidity has synergistic bidirectionality, meaning that headache increases the burden of TMD and TMD increases the burden of headache [6,14], thus increasing the risk of developing persistent pain. Chronic pain has many consequences on physical, psychological, and social functioning, so the patients’ cognitive content, pain-related behaviors, health care demands, and response therapies are highlighted as relevant factors for aggravation or pain maintenance [17].

We have found that being a middle-aged female, having lesser education, and having family members presenting headache are all linked to a higher probability of reporting chronic pain from headache and TMD. Our results are in accordance with previous studies [48,49,50] and may be attributed to age, gender, social status, and education, among other socioeconomic factors, which are known to influence health self-perception [9,51,52,53,54]. In addition, there is evidence that females have characteristics that make them more prone to chronic pain, such as hormones (estrogen levels) and genetics [55]. There are also psychological traits that may explain the fact that females use more maladaptive coping mechanisms, predisposing them to chronic pain and reduced functional abilities [56].

Low education and poor general health are two socio-demographic characteristics associated with oral discomfort [48,51,57]. Chronic pain is more common in those with low levels of education, perceived financial inequities, and significant levels of local impoverishment [58] which suggests that lower education can obstruct the practice of healthy behaviors and the assigned value on health in general, and oral health in particular [59]. 

When referring to onset or perpetuating variables of TMD and headache, it is important to mention that the literature shows no connection between TMD/headache and orthodontic treatment. In addition, there is inadequate data to support the idea that occlusal adjustment or orthodontic treatment of malocclusion can help prevent TMD [60,61,62].

We also verified that the TMD-headache group when compared to the headache group is the one with the highest values in the three dimensions of psychological inflexibility: pain avoidance, cognitive fusion, and inflexibility in pain (total). Our findings are thus in accordance with the results from previous studies that found that patients with comorbid physical and mental chronic diseases are more liable to suffer chronic pain than those without such diseases [63,64].

Patients with chronic pain typically put up a great deal of effort to combat their pain, which includes not just physical sensations but also emotions, memories, and ideas regarding pain [17]. Our findings, as previously stated in existing literature [38,65], show that individuals suffering from both these clinical conditions tend to show greater pain intensity, resulting mostly in a greater psychological inflexibility in pain.

Psychological flexibility increases the capacity to adapt behaviors according to the situation at hand, and to pursue central interests and long-term goals rather than narrowly concentrating on achieving happiness or short-term goals [66,67]. For instance, chronic pain patients with high psychological flexibility tend to realize that pain should not prevent activity and so they tend to continue to do what they value, unaffected by painful experiences [24].

Our results have shown that suffering from both these clinical conditions seems to cause greater pain inflexibility, resulting in more vivid suffering experiences that lead to altering daily decisions and actions. In such patients those negative feelings and sensations will likely lead to a reduction of physical activity and avoiding social contact and leisure [50]. Furthermore, in the short term, avoiding discomfort is frequently followed by a reduction in unpleasant stimuli; hence, these behaviors are negatively reinforced [68]. Emshoff et al., 2017, also verified that both conditions have been linked to high levels of depression, somatization, and pain-related disability [69]. The bidirectional relationship between TMD and headache comorbidity may be explained by the shared nociceptive system [6,70]. The first neurons engaged in headache are linked to the trigeminocervical complex and the first branch of the trigeminus, whereas those involved in TMD are connected to neurons in the first and second branches of the trigeminus. This nociceptive input converges on the trigeminus caudal nucleus, and from there, headache and TMD share unique core pathways [6,11,39,71]. Having this, we may hypothesize that individuals suffering from both conditions will also tend to accumulate the difficulties of pain management inherent to each of those conditions which should be considered in new treatment protocols.

### 4.2. Pain Triggers in the Fear-Avoidance Model

In our headache group, we verified that higher pain intensity and lower availability to pain contribute to a higher psychological inflexibility in pain. Those findings are in line with Lillis et al., 2017 who concluded that higher levels of pain willingness (a subcomponent of pain acceptance) were associated with lower levels of headache-related disability and pain interference with daily functioning [72].

In our TMD-headache group, what contributed to pain inflexibility was the perception of anxiety and nervousness, and also pain intensity. People with headache usually describe a wide range of perceived precipitants or triggers [73,74], and avoidance of perceived triggers in daily activities due to fear of pain [75]. Behavioral avoidance due to fear of pain can result in a lowering of the pain threshold, failed habituation to pain, and a lack of opportunity for learning to cope with it, ultimately leading to disability [76,77]. This fear-avoidance model has been well established in chronic musculoskeletal pain [78] and more recently in headache-related pain [75,79]. Psychological variables, such as anxiety sensitivity—the general tendency to interpret somatic symptoms as aversive or dangerous—appear to play a larger role in exacerbating the fear-avoidance relationship than headache symptoms, such as headache severity, Ref. [80] making it possible to intervene in the fear-avoidance relationship via psychosocial treatment aiming to improve outcomes (as studied on migraine patients [72]).

The model that studied the role of illness years as a mediator in the link between pain willingness and psychological inflexibility in the headache group showed significant results, meaning that when the sickness years variable was high, there was a negative connection between pain willingness and psychological inflexibility, demonstrating that longer sickness years result in stronger psychological inflexibility and lower pain acceptance. These results could be explained by the individual attempts to change internal triggers or avoid them over the years. Long-term, excessive avoidance of all triggers (particularly internal ones such as stress, pain, undesirable thoughts, feelings, and emotions) regardless of context becomes counterproductive, resulting in additional suffering, paradoxically increasing the experiences that one is seeking to avoid. This was also concluded by Vasiliou et al., 2021, which described that trigger avoidance increases trigger potency, reduces pain tolerance, limits lifestyle, reduces internal locus of control, and worsens headache [68].

### 4.3. Pain Management of TMD-Headache

The frequency of TMD, but also its severity, is linked to headache, validating the hypothesis that patients suffering from both conditions are more likely to have complicated and severe TMD diagnoses as well as increasing headache variables including frequency, duration, and intensity, as a result of overlapping symptoms and the continual presence of pain [69,81,82]. This increased pain sensation, when suffering from both conditions, can be explained, following the pain adaptation theory and its conclusion that different types of pain tend to reinforce each other [1].

In epidemiologic and clinical studies, variation in emotional function has been linked to the severity of both TMD and headache [51,69,83]. As a result, stress must be diagnosed and adequately managed. The initial approach for TMD treatment should be reversible and non-invasive [84]. Therapy will vary according to the clinical characteristics of each individual [5,85]; therefore, it is advisable to set a customized treatment plan with a minimal side effect profile [86] which may include self-management, psychological interventions, physical therapy, pharmacotherapy, and occlusal splint therapy [84,85].

To enhance well-being and address the psychopathological symptoms of TMD, psychological follow-up is essential [5,23]. Psychological therapies, such as cognitive-behavioral therapy, aim to reduce depression symptoms, anxiety, stress, and pain intensity in people with TMD [87]. Innovative non-pharmacological approaches, such as acceptance-based treatments like acceptance and commitment therapy (ACT), are studied and applied with excellent empirical evidence in a variety of medical disorders, including chronic pain [88,89]. The basic purpose of ACT is to increase psychological flexibility, increasing the capacity to respond to changes in the environment, allowing the patient to feel and think about what is happening at a given time [89], working through six related points of interest: acceptance, cognitive fusion, mindfulness, self-observance, values and committed actions [68,89]. There are, however, different factors that should be considered when analyzing our findings. One aspect is that, even though we have controlled the medication intake of the participants, it was not our objective to analyze who prescribed it, with what aims they prescribed it, nor its influence on pain perception. However, because we acknowledge that medication intake changes pain perception and that self-medication is often seen in clinical practice, future studies, with higher sample sizes, should try to address and control these aspects. Also, we have used a non-probabilistic sample with a convenience sample and this may be considered a limitation to our findings; in fact, a random sample of community members would have provided more generalizable results. Sociodemographic variables and clinical variables, like the ones used in this study, are the base to understand and interpret chronic pain conditions like headache and TMD.

It should be noted that we have used DC/TMD for the diagnosis of TMD, and this instrument does not encompass the occlusal status of the individual. Although studies have shown that occlusion is important not only for the development of TMD but also for its treatment [90,91,92], others stated that different types of malocclusion might be linked to headache [12,14,62,90,91,92,93,94]. However, most of the literature does not consider these factors as major aspects in TMD aetiology, but they cannot be completely dissociated [60,61,62,93,94].

There is a need for a multidisciplinary approach, creating a broad treatment strategy. This strategy should focus on both clinical conditions, and the complicating factors associated. Importance should be given to the biopsychosocial component—focusing on lowering psychological inflexibility and increasing acceptance of pain. Future studies should integrate psychological variables as premorbid risk factors for the development of other chronic pain conditions, including chronic widespread pain, and musculoskeletal pain. It would also be interesting to research the efficacy of psychological therapies such as ACT in patients with both TMD and headache, since this is a successful treatment for headache and other diseases with chronic pain. Continued research in this field should improve our understanding of the complex relationship between headache and TMD, as well as how innovative behavioral methods might improve treatment efficacy, resulting in better management of these debilitating and hard-living illnesses. Psychopathological symptoms such as anxiety and depression should be assessed as part of the clinical examination of patients with one or both conditions, since it is critical to understand these and other variables in patient management.

## 5. Conclusions

Our results have shown that the higher the psychological inflexibility, the lower the pain acceptance, showing that the presence of TMD and/or headache leads the individual to have more difficulty controlling their pain, and that patients with both conditions are more prone to chronic pain and pain inflexibility.

In the headache group, we found out that being a male, having no systemic disease, having less pain severity but a higher frequency of pain, living longer with the disease, and having sensitive changes, showed higher pain acceptance.

When headache occurs with TMD, females with higher education, no headache family history, less pain, and no motor changes showed higher pain acceptance. New interventions can now have empirical elements to sustain detailed manualization, envisaging higher efficacy in treating TMD and headache patients.

## Figures and Tables

**Figure 1 ijerph-19-07974-f001:**
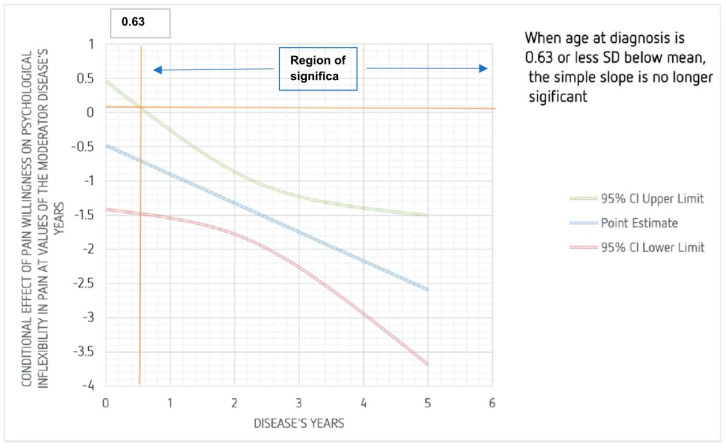
Moderating role of illness years in the relation between pain willingness and psychological inflexibility in pain in headache group.

**Table 1 ijerph-19-07974-t001:** Sample’s sociodemographic and clinical characteristics: differences between groups.

		Total	Control	Headaches	TMD-Headache	Differences		
**Sociodemographic Variables**		***n* (%)**	***n* (%)**	***n* (%)**	***n* (%)**	** *χ^2^* **	** *p* **	**φ**
Sample		120 (100.0)	25 (20.83)	61 (50.83)	34 (28.33)			
Gender	Female	80 (66.7)	13 (52.0)	40 (65.6)	27 (79.4)	4.94	0.085	0.20
	Male	40 (33.3)	12 (48.0)	21 (34.4)	7 (20.6)			
Marital status	No relation	49 (40.8)	9 (36.0)	27 (44.3)	13 (38.2)	0.63	0.728	0.07
	In a relation	71 (59.2)	16 (64.0)	34 (55.7)	21 (61.8)			
Education	Without university studies	78 (65.0)	17 (68.0)	40 (65.6)	21 (61.8)	0.264	0.876	0.05
	With university studies	42 (35.0)	8 (32.0)	21 (34.4)	13 (38.2)			
						** *F* **	** *p* **	**η^2^**
Age	*M* ± *SD*; Min–Max	46.89 ± 16.64; 10–89	52.00 ± 19.12; 18–89	45.92 ± 15.70; 10–72	44.88 ± 16.09; 15–71	1.54	0.218	0.03
**Clinical variables**						** *χ^2^* **	** *p* **	**φ**
Systemic disease	No	57 (47.5)	9 (36.0)	32 (52.5)	16 (47.1)	1.93	0.381	0.13
	Yes	63 (52.5)	16 (64.0)	29 (47.5)	18 (52.9)			
Medication	No	40 (33.3)	7 (28.0)	25 (41.0)	8 (23.5)	3.40	0.183	0.17
	Yes	80 (66.7)	18 (72.0)	36 (59.0)	26 (76.5)			
Headaches	No	22 (18.3)	21 (84.0)	1 (1.6)	0 (0.0)	90.99	<0.001	0.87
	Yes	98 (81.7)	4 (16.0)	60 (98.4)	34 (100.0)			
Attendance to doctor	Not applicable	22 (18.3)	21 (84.0)	1 (1.6)	0 (0.0)	91.01	<0.001	0.87
	No	43 (35.8)	2 (8.0)	26 (42.6)	15 (44.1)			
	Yes	55 (45.8)	2 (8.0)	34 (55.7)	19 (55.9)			
Illness duration	Not applicable	31 (25.8)	21 (84.0)	4 (6.6)	6 (17.6)	63.29	<0.001	0.73
	≤20 years	62 (51.7)	3 (12.0)	35 (57.4)	24 (70.6)			
	>20 years	27 (22.5)	1 (4.0)	22 (36.1)	4 (11.8)			
Headaches and Anxiety/Stress/Nervousness	Not applicable	31 (25.8)	21 (84.0)	10 (16.4)	0 (0.0)	59.70	<0.001	0.71
	No	54 (45.0)	2 (8.0)	33 (54.1)	19 (55.9)			
	Yes	35 (29.2)	2 (8.0)	18 (29.5)	15 (44.1)			
Family with headache	Not applicable	9 (7.5)	8 (32.0)	1 (1.6)	0 (0.0)	28.40	<0.001	0.49
	No	12 (10.0)	2 (8.0)	5 (8.2)	5 (14.7)			
	Yes	99 (82.5)	15 (60.0)	55 (90.2)	29 (85.3)			
Pain severity	Not applicable	23 (19.2)	21 (84.0)	2 (3.2)	0 (0.0)	87.75	<0.001	0.86
	Light	37 (30.8)	0 (0.0)	24 (39.3)	13 (38.2)			
	Moderate	28 (23.3)	3 (12.0)	17 (27.9)	8 (23.5)			
	Severe	32 (26.7)	1 (4.0)	18 (29.5)	13 (38.2)			
						** *F* **	** *p* **	**η^2^**
Pain intensity	*M* ± *SD*; Min–Max	5.82 ± 3.45; 0–10	1.24 ± 2.96; 0–10	7.10 ± 2.34; 0–10	6.91 ± 2.54; 2–10	51.79	<0.001	0.47
						** *χ^2^* **	** *p* **	**φ**
Pain frequency	Not permanent	32 (27.7)	21 (84.0)	7 (11.5)	4 (11.8)	53.08	<0.001	0.67
	Permanent	88 (73.3)	4 (16.0)	54 (88.5)	30 (88.2)			
						** *F* **	** *p* **	**η^2^**
Pain years	*M* ± *SD*; Min–Max	21.22 ± 16.61; 0–60	21.50 ± 24.34; 0–60	21.38 ± 15.92; 1–55	20.91 ± 17.44; 0–60	0.01	0.991	0.00
						** *χ^2^* **	** *p* **	**φ**
Medical exam	Not applicable	22 (18.3)	21 (84.0)	1 (1.6)	0 (0.0)	91.07	<0.001	0.87
	No	68 (56,7)	3 (12.0)	41 (67.2)	24 (70.6)			
	Yes	30 (25.0)	1 (4.0)	19 (31.1)	10 (29.4)			
Previous treatment	Not applicable	22 (18.3)	21 (84.0)	1 (1.6)	0 (0.0)	91.31	<0.001	0.87
	No	21 (17.5)	0 (0)	13 (21.3)	8 (23.5)			
	Yes	67 (64.2)	4 (16.0)	47 (77.1)	26 (76.5)			
Preventive treatment	Not applicable	22 (18.3)	21 (84.0)	1 (1.6)	0 (18.3)	91.01	<0.001	0.87
	No	78 (65.0)	3 (12.0)	48 (78.7)	27 (79.4)			
	Yes	20 (16.7)	1 (4.0)	12 (19.7)	7 (20.6)			
Visual changes	Not applicable	22 (18.3)	21 (84.0)	1 (1.6)	0 (0.0)	91.06	<0.001	0.87
	No	56 (46.7)	2 (8.0)	34 (55.7)	20 (58.8)			
	Yes	42 (35.0)	2 (8.0)	26 (42.6)	14 (41.2)			
Sensitive changes	Not applicable	22 (18.3)	21 (84.0)	1 (1.6)	0 (0.0)	91.58	<0.001	0.87
	No	90 (75.0)	3 (12.0)	55 (90.2)	32 (94.1)			
	Yes	8 (6.7)	1 (4.0)	5 (8.2)	2 (5.6)			
Motor changes	Not applicable	22 (18.3)	21 (84.0)	1 (1.6)	0 (0.0)	91.30	<0.001	0.87
	No	91 (75.8)	4 (16.0)	55 (90.2)	32 (94.1)			
	Yes	7 (5.8)	0 (0)	5 (8.2)	2 (5.6)			
Anxiety/nervousness causes headaches	Not applicable	21 (17.5)	20 (80.0)	1 (1.6)	0 (17.5)	85.69	<0.001	0.85
	No	46 (38.6)	2 (8.0)	29 (47.5)	15 (44.1)			
	Yes	53 (44.2)	3 (12.0)	31 (50.8)	19 (55.9)			
Anxiety/nervousness aggravates headaches	Not applicable	21 (17.5)	20 (80.0)	1 (1.6)	0 (17.5)	93.37	<0.001	0.88
	No	58 (56.7)	3 (12.0)	47 (77.0)	18 (52.9)			
	Yes	31 (25.8)	2 (8.0)	13 (21.3)	16 (47.1)			
Headache diagnosis	No	25 (20.8)	25 (100.0)	0 (00.0)	0 (0.0)	120.00	<0.001	1.00
	Yes	95 (79.2)	0 (0.0)	61 (100.0)	34 (100.0)			

*M* = mean; *SD* = standard deviation; Min = Minimum; Max = Maximum; *F* = F-test in One-Way ANOVA; *p* = *p*-value; η^2^ = eta squared (size effect); *χ^2^* = qui-squared; φ = Phi = (size effect).

**Table 2 ijerph-19-07974-t002:** Descriptive statistics for psychological variables in pips and CPAQ-8 scales.

	Total	Control	Headache	TMD-Headache	Differences		
	*M* (*SD*)	*M* (*SD*)	*M* (*SD)*	*M* (*SD*)	*F*	*p*	η^2^
Psychological Inflexibility in Pain Scale (PIPS) Total	35.86 (12.18)	30.72 (20.16) ^1^	32.10 (12.23) ^2^	46.38 (18.51) ^1,2^	6.43	0.002	0.10
PIPS Avoidance of Pain	20.08 (13.43)	17.32 (12.96) ^1^	18.11 (12.77) ^2^	25.62 (13.68) ^1,2^	4.30	0.016	0.07
PIPS Pain Cognitive Fusion	15.78 (9.22)	13.40 (8.42) ^1^	13.98 (9.52) ^2^	20.76 (7.36) ^1,2^	7.76	0.001	0.12
Chronic Pain Acceptance Questionnaire (CPAQ) Total	25.35 (5.69)	24.96 (3.51)	25.34 (6.57)	25.65 (5.40)	0.10	0.902	0.00
CPAQ Activity Engagement	14.12 (9.42)	13.56 (10.68)	13.03 (9.78)	16.47 (7.39)	1.52	0.222	0.03
CPAQ Pain Willingness	11.23 (9.09)	11.40 (10.19)	12.31 (9.41)	9.18 (7.40)	1.31	0.273	0.02

*M* = mean; *SD* = standard deviation; *F* = F-test in One-Way ANOVA; *p* = *p*-value; η^2^ = eta squared (size effect); ^1,2^ = Tuckey HSD.

**Table 3 ijerph-19-07974-t003:** Variables that contribute to psychological inflexibility in pain in headache group.

Variables	Model 1			Model 2		
	* **B** *	*SE B*	*ß*	* **B** *	*SE B*	*ß*
Pain intensity	4.33	1.04	0.48	3.55	0.84	0.39
Pain willingness				−1.23	0.21	−0.55
*R*^2^(*R*^2^Aj.)	0.23 (0.21)			0.52 (0.50)		
*F* for change in *R*^2^	17.31 ***			35.05 ***		

Note. *B* = unstandardized regression coefficients; *ß* = standardized regression coefficients; *** *p* < 0.001.

**Table 4 ijerph-19-07974-t004:** Variables that contribute to psychological inflexibility in pain in TMD-headache group.

Variables	Model 1		
	* **B** *	*SE B*	*ß*
Pain intensity	3.82	1.02	0.52
Anxiety/nervousness that causes headache	10.94	5.11	0.30
*R*^2^(*R*^2^Aj.)	0.47 (0.44)		
*F* for change in *R*^2^	13.84 ***		

Note. *B* = unstandardized regression coefficients; *ß* = standardized regression coefficients; *** *p* < 0.001.

**Table 5 ijerph-19-07974-t005:** Variables that contribute to pain acceptance in headache group.

Variables	Model 1			Model 2		
	* **B** *	*SE B*	*ß*	* **B** *	*SE B*	*ß*
Gender (Male)	−3.25	1.76	−0.24	−5.78	1.69	−0.42
Systemic disease				−4.89	1.50	−0.37
Pain severity				−2.22	0.83	−0.31
Pain frequency				5.87	2.68	0.28
Illness duration				0.10	0.05	0.24
Sensitive changes				4.12	1.62	0.31
*R*^2^(*R*^2^Aj.)	0.06 (0.04)			0.35 (0.28)		
*F* for change in *R*^2^	3.42 (*ns*)			4.83 **		

Note. *B* = unstandardized regression coefficients; *ß* = standardized regression coefficients; ** *p* < 0.001; *ns* = not significant.

**Table 6 ijerph-19-07974-t006:** Variables that contribute to pain acceptance in the TMD-headache group.

Variables	Model 1			Model 2		
	* **B** *	*SE B*	*ß*	* **B** *	*SE B*	*ß*
Gender (Female)	2.65	2.20	0.20	5.44	2.23	0.41
Education	0.67	0.39	0.29	0.81	0.35	0.35
Family with headache				−5.73	2.62	−0.38
Pain frequency				−5.26	2.54	−2.07
Motor changes				−8.43	3.59	−0.37
*R*^2^(*R*^2^Aj.)	0.13 (0.07)			0.38 (0.27)		
*F* for change in *R*^2^	2.30 (*ns*)			3.77 *		

Note. *B* = unstandardized regression coefficients; *ß* = standardized regression coefficients; * *p* < 0.001; *ns* = not significant.

## Data Availability

The authors declare that the data supporting the findings of this study are available within the article.

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
