# Peer review of "The Role of Pain Inflexibility and Acceptance among Headache and Temporomandibular Disorders Patients"

_ijerph, 2022, doi:10.3390/ijerph19137974_

Round 1
Reviewer 1 Report
The paper entitled "THE ROLE OF PAIN INFLEXIBILITY AND ACCEPTANCE AMONG HEADACHE AND TEMPOROMANDIBULAR DISORDERS PATIENTS" is an original article aiming to assess the association between TMD, headache, psychological dimensions and consequences on life quality, pain, and emotional regulation.
The grammar is unclear on some points and needs revision by native speakers or specific tools.
Abstract: the "Methods" lack information about the enrollment of subjects.
Line 166 "When it comes to the clinical variables, half of our sample presented a systemic disease (52.5%), 66.7% takes medication". Is it possible to report the medications (for systemic diseases or TMD/headaches) and diseases?
Tables: while the significant p-value in table 1 was set and reported as "<0.001", in table 2, it was reported as the precise value in bold type ("0.002…"). Please, uniform how you intend to present the data in all tables.
Table 5 and 6: please, specify in a caption to the table what gender contributes to pain acceptance in both groups, as reported in the text.
Not mandatory, I suggest adding the complete questionnaire adopted in English as supplementary material for a better fruition of the content by the interested readers.
In discussion, in lines 267-269 authors stated, "we have found that being a middle-aged woman, having lesser education, […] are all linked to a 268 higher probability of reporting chronic pain from headaches and TMD". Could the lesser education and the lack of dental/orthognathic treatment be a variable to consider in the onset of headache/TMD?
In this case, the authors should have considered the presence of malocclusion among those suffering headaches and/or TMD. Please, briefly discuss the limitation of the study of not considering the occlusal status or the etiology of the TMD. I mean, despite the "psychological" point of view of the paper and the suggestions to consider a psychological approach to reduce the pain sensation, the treatment of the clinical condition by appropriate therapy must be considered and reported as the first line of treatment.
Along with the text, the authors did not consider the use of pain relief drugs and their side effects on prolonged administration. Furthermore, many subjects did not have medical exams and could self-treat the pains incorrectly. Similarly, when referring to "previous treatments" and "preventive treatments", authors should clarify what kind of treatment (pharmacological, T.E.N.S. and so on..) they received and who prescribed with what results in pain relief.
Reviewer 2 Report
This study analyses the relationship between Temporomandibular disorders (TMD) and headaches , psychological dimensions and consequences on life quality, pain and emotional regulation by using certain criteria: Diagnostic Criteria for Temporomandibular Disorders (DC-TMD), International Classification of Headache Disorders (ICHD-3 beta version), Chronic Pain Acceptance Questionnaire (CPAQ-8) and Pain Flexibility Scale Psychological (PIPS).
Although an original and intriguing subject the manuscript has several issues that must be addressed.
Please see the enclosed PDF for further details.

Reviewer 3 Report
The text should be reviewed by a proficient English speaker.
The abstract is not in the format required by the journal (please remove headings).
Please check the references format in the journal template.
The article is extremely verbose, but the novelty or the significance of the results are not easy to grasp from the text.
Also the conclusion is rather general.
Round 2
Reviewer 3 Report
I am satisfied with the changes.